# Influence of Carbon Black and Silica Fillers with Different Concentrations on Dielectric Relaxation in Nitrile Butadiene Rubber Investigated by Impedance Spectroscopy

**DOI:** 10.3390/polym14010155

**Published:** 2021-12-31

**Authors:** Gyung-Hyun Kim, Young-Il Moon, Jae-Kap Jung, Myung-Chan Choi, Jong-Woo Bae

**Affiliations:** 1Department of Physics and Research Institute of Natural Science, Gyeongsang National University, Jinju 52828, Korea; 2016010673@gnu.ac.kr; 2Department of Electrical Engineering, Pohang University of Science and Technology, Pohang 37673, Korea; yimoon@postech.ac.kr; 3Hydrogen Energy Materials Research Center, Korea Research Institute of Standards and Science, Daejeon 34113, Korea; 4Rubber Research Division, Korea Institute of Footwear & Leather Technology, Busan 47154, Korea; mcchoi@kiflt.re.kr (M.-C.C.); jwbae@kiflt.re.kr (J.-W.B.)

**Keywords:** nitrile butadiene rubber, dielectric relaxation, impedance spectroscopy, carbon black, activation energy

## Abstract

In neat nitrile butadiene rubber (NBR), three relaxation processes were identified by impedance spectroscopy: α and α′ processes and the conduction contribution. We investigated the effects of different carbon black (CB) and silica fillers with varying filler content on the dielectric relaxations in NBR by employing a modified dispersion analysis program that deconvolutes the corresponding processes. The central frequency for the α′ process with increasing high abrasion furnace (HAF) CB filler was gradually upshifted at room temperature, while the addition of silica led to a gradual downshift of the center frequency. The activation energy behavior for the α′ process was different from that for the central frequency. The use of HAF CB led to a rapid increase in DC conductivity, resulting from percolation. The activation energy for the DC conductivity of NBRs with HAF CB decreased with increasing filler, which is consistent with that reported in different groups.

## 1. Introduction

Nitrile butadiene rubber (NBR) is the most commonly utilized elastomer in the automotive, aeronautical, and nuclear industries [1,2]. NBR resists oil-based hydraulic fluids, vegetable oils, flame retardant liquids, grease, water, and gas [3,4]. Although NBR is known to exhibit numerous outstanding properties, enhancement fillers are necessarily added to NBR to attain appropriate properties for specific applications, such as low gas permeation at high pressure. The reinforcement of elastomers improves physical properties such as tear strength, tensile strength, hardness, abrasion resistance, and thermal properties. A wide variety of particulate fillers are used in the rubbery polymer industry for various purposes, of which the most important are reinforcement, reduction in material costs, and improvements in processing [5,6]. Physical properties such as volume swelling, density, chemical stability, and heat resistance of rubber vulcanizates are also improved through incorporation with fillers such as carbon black (CB) [7] and silica [8].

CB and silica have been used as the main reinforcing agents in rubber compounds, but their surface chemistries are very different [9]. CB is one of the most widely used conductive nanoparticles in industrial applications and is considered a suitable candidate because of its low cost and good dispersion ability. When CB is compounded with rubbers, the tensile strength, tear strength, modulus, and abrasion resistance are found to be increased [10,11]. For this reason, CB has been extensively exploited in numerous rubber engineering products [12]. On the other hand, silica provides a unique combination of tear strength, abrasion resistance, aging resistance, and adhesion properties [13]. In tire treads, silica yields a lower rolling resistance at equal wear resistance and wet grip than CB [14]. Synthetic silica is currently extensively used to improve physical and mechanical parameters, such as tensile strength and elongation of silicone rubber vulcanisates [12]. The use of silica fillers enables the physical and mechanical properties of NBR rubber vulcanisates to be improved [15,16].

Under these circumstances, it is necessary to research the effects of fillers on these properties in rubber composites by employing a non-destructive microscopic technique. Impedance spectroscopy provides useful ways to clarify the dielectric relaxation and related electrical properties of rubber composites. Thus, it has been extensively employed to investigate the nature of local molecular motion associated with relaxation behavior [17,18,19,20,21]. However, limited investigations of the influence of fillers on dielectric properties in NBR have taken place thus far [22,23]. It is a crucial task to provide useful information on dielectric responses of NBRs blended with various filling agents in rubber applications and related academic studies. Therefore, the main focus of this investigation lies in the influences of the various fillers on the dynamic relaxation properties of NBR.

Recently, we characterized several relaxation processes embedded with NBR, ethylene propylene diene monomer (EPDM), and fluoroelastomers (FKM) with a self-developed dispersion analysis program by employing the modified numerical method of nonlinear optimization [24,25]. As a continuation of that line of research, the present investigations are concerned with systematic studies of the electrical properties of blends of neat NBR mixed with CB and silica. The effect of filler loading on the dielectric relaxation process and DC conductivity was studied in these composites in an attempt to understand the corresponding mechanism and filler-induced effects. Thus, the present paper addresses conductive rubber based on blends of NBR in different proportions and filled with different amounts of conductive CB and silica. The effect of the filler on the α and α’ processes and conductivity are discussed in terms of the corresponding process motion, activation energy, and glass transition.

## 2. Materials and Methods

### 2.1. Sample Composition

KNB 35 L (Kumho NBR) with an acrylonitrile content of 34 wt%, produced by Kumho petrochemical group, was used as the main component for neat NBR rubber. The compound recipe for the composition for NBR specimens with CB and silica fillers is given in Table 1 and Table 2, respectively, which includes one neat NBR without any added filler, six samples with CB, and three samples with silica filler. In this study, we employed two types of CB prepared using a high abrasion furnace (HAF) and a medium thermal furnace (MT) by Orion Engineer Carbon, which have particle sizes of 28–36 nm and 250–350 nm, respectively. The detailed ASTM (American Society for Testing and Materials) classifications of our CB particles are listed in Appendix A. The silica (specific grade Zeosil^®^ 175) was produced by Solvay, which has a specific surface area of 175 m^2^/g.

The vulcanizates were filled with 20 phr, 40 phr, and 60 phr (parts per 100 parts of rubber). For simplicity, the NBR blends mixed with fillers were named NBR-Hx, NBR-My, and NBR-Sz, where x, y, and z indicate the phr content for HAF, MT, and silica, respectively. For example, NBR-S40 is NBR filled with silica of 40 phr.

A two-stage mixing was employed, using the internal mixer with two banbury rotors and two open roll mills of eight inches to prepare NBR composites. The first stage of mixing (masterbatch) was compounding of NBR rubber, reinforcing the fillers such as carbon blank and precipitated silica, and processing the aids such as ZnO and the stearic acid with an internal mixer (3L kneader, Moriyama Co., Tokyo, Japan). The filling factor was fixed to 0.8, and the starting operation temperature of kneader was set to 80 °C. The rotor speed was set to 30 rpm. The NBR rubber was added to 3L kneader and masticated for 3 min. After this, the reinforcing filler and the processing aids were incorporated for 10 min. At the second stage of mixing, open roll mills were used to add the curing agents and accelerating agents into the masterbatch composite. The mixer was set to a nip opening of 3 mm between the rolls. The masterbatch was added to the roller and mixed for 1 min. The sulfur and TBBS were then added and mixed into the batch, which took about 2 min. The mixer nip was opened, and then the finished batch was cut into sheets. The mixing time was kept uniform for all composites.

Vulcanizate sheets of the composites with a thickness of 3 mm for impedance testing were prepared by compression molding in a hydraulic press at 150 °C according to the optimum cure time obtained from the oscillating disk rheometer.

### 2.2. Impedance Spectroscopy

The impedance spectroscopy system includes a temperature chamber, an impedance analyzer (VSP-300 BioLogic, Seyssinet-Pariset, France), and a PC with a dispersion analysis program for controlling the temperature chamber and processing the measured impedance data. The configurations of the system are well described in previous papers [24,25], and the description of the analysis software is found in Appendix A. All of the specimens were prepared with a cylindrical shape with a diameter of 70 mm and a thickness of 3 mm, and both top and bottom surfaces were covered by the copper electrodes.

The impedance measurement was conducted in the frequency range of 0.01 Hz to 1 MHz and a temperature range of 233 K to 403 K over 5 K intervals at an applied voltage of 1000 mV. The real and imaginary dielectric permittivity was obtained from the impedance measurement by the following relation:(1)$$\varepsilon^{\prime }=\frac{d}{\omega \varepsilon_{0}A}\frac{Z^{\prime \prime }}{\left({Z^{\prime }}^{2}+{Z^{\prime \prime }}^{2}\right)}\;,\;\;\;\varepsilon^{\prime \prime }=\frac{d}{\omega \varepsilon_{0}A}\frac{Z^{\prime }}{\left({Z^{\prime }}^{2}+{Z^{\prime \prime }}^{2}\right)}$$
where $Z^{\prime }$ and $Z^{\prime \prime }$ are the real and imaginary parts of the impedance, respectively. d is the thickness of the sample, A is the area of the sample, and *ω* is the angular frequency. $\varepsilon_{0}$ is the vacuum permittivity, $\varepsilon_{0}$ = 8.854 × 10^−12^ F/m.

### 2.3. Transmission Electron Microscopy and Differential Scanning Calorimeter

The microstructure of NBR samples was investigated with a combination of the focused ion beam (FIB) and transmission electron microscopy (TEM). Thin foil specimens for TEM observation were prepared by a FIB technique. Morphology, distributions, and size of the filler particles in NBR specimens were observed with a TEM (TECHNAI F20, FEI company) operated at an accelerating voltage of 200 kV.

Dynamic glass transition temperature (T_g_) for NBR specimens was measured by using a differential scanning calorimeter (DSC, Q-600 analyzer of TA instruments). The measurements were conducted with a heating rate of 1 K/min in the temperature range from 193 K to 353 K. The difference in heat flow (W/g) into the sample and the reference pans were recorded as a function of temperature. From the measured heat flow curves, T_g_ was calculated by TA universal analysis software, a program that was also provided by TA instruments.

## 3. Relaxation Process and Model Function

### 3.1. α Relaxation Process

The α relaxation process is mainly related to the segmental motion of the amorphous polymer chains and glass transition. The α relaxation observed normally at lower temperatures (250 K–300 K) and high frequencies (10^6^–10^10^ Hz) has a strong temperature dependence with non-Arrhenius characteristics. It can be described as a Vogel–Fulcher–Tammann–Hesse (VFTH) function:(2)$$f_{0}=f_{\infty \;}\exp\left[-\frac{E_{a}}{k_{B}\left(T-T_{0}\right)}\right]$$
where $f_{0\;}$ is the central frequency of the α process in the frequency domain, $f_{\infty \;}$ is a constant, and *k_B_* is the Boltzmann constant. *T* is the temperature, and *T*_0_ is the so-called ideal glass transition or Vogel temperature, which is normally found to be 30–70 K below the glass transition temperature (*T_g_*) [19].

### 3.2. α′ Relaxation Process

The α′ relaxation process contributes to the motion of the end-to-end vector in polymer segmental chains, which is observed only in polymer type A. According to Stockmayer [26] and Block [27], the vector sum of the total dipole moment in polymer chains consists of the parallel direction with the segmental chain (backbone chain), called polymer type A. For this polymer type, when proportional to the fluctuation of the end-to-end vector of the polymer chain, α′ relaxation can be observed. α′ relaxation is usually called normal mode relaxation. Similar to α relaxation, the central frequency of α′ relaxation has nonlinear temperature dependence, and the relaxation rate of α′ can be described by the VFTH temperature dependence.

### 3.3. Conductivity and Model Function

The origin of DC conductivity in polymers can be regarded as a transport phenomenon by considering the movement of the charge carriers such as ionic impurities [28] and thermally excited electrons between valance and conduction band [29]. In this study, the measurement of the DC conductivity using dielectric spectroscopy was obtained by the relation between Maxwell equation and Jonscher’s power law given by
(3)$$\nabla \times H=\left(\sigma_{tot}+j\omega \varepsilon \right)E=j\omega \varepsilon \left(1-j\frac{\sigma_{tot}}{\omega }\right)E=j\omega \varepsilon^{*}E$$
(4)$$\sigma_{tot}=\sigma_{dc}+A\omega^{n}$$
where **H** and **E** indicate the magnetic and electric field vectors, respectively. The $\sigma_{tot}$ is the total conductivity of the sample, $A\omega^{n}$ is the pure dispersive component of AC conductivity with a characteristic of power law in terms of angular frequency ω and exponent n (0≤n≤1), and A is a constant that determines the strength of polarizability [30]. The temperature dependency of DC conductivity ($\sigma_{dc})\;$is described by the Arrhenius equation:(5)$$\sigma_{dc}=\sigma_{0}\exp\left[-\frac{E_{a}}{k_{B}T}\right]$$
where σ_0_ is the pre-exponential factor and *E_a_* is the activation energy.

In investigating the dynamics of the corresponding relaxation process in polymers, the experimental complex dielectric permittivity $\varepsilon^{*}\left(\omega \right)$ data can be fitted to the Havriliak–Nagami (HN) model, which can be expressed as a combination of the conductivity term and several HN functions forms as follows [18,19,31,32]:(6)$$\varepsilon^{*}\left(\omega \right)=-i{\left(\frac{\sigma_{dc}}{\varepsilon_{0}\omega }\right)}^{N}+\varepsilon_{\infty }+\underset{k=1,2,\ldots }{\sum }\frac{\Delta \varepsilon_{k}}{{\left[1+{\left(i\omega \tau_{HN\;k}\right)}^{1-a}\right]}^{b}}$$
where the first term ${\left(\frac{\sigma_{dc}}{\varepsilon_{0}\omega }\right)}^{N}$ accounts for the conductivity, and *N* is an exponent that characterizes the conduction process. The conductivity contribution has the strongest influence at the highest temperature and lowest frequency. The last term is the HN function. The summation symbol indicates the occurrence of more than one relaxation process. $\tau_{HN\;k}$ is the average characteristic relaxation time of the corresponding process *k*. $\Delta \varepsilon_{k}$ is the dielectric relaxation strength or intensity. For the fractional shape parameters, *a* and *b* describe the linewidth and asymmetry of the relaxation loss peak, respectively, where *a* ≥ 0 and *b* ≤ 1 hold. When *a* = 1 and *b* = 1, the linewidth of the relaxation process becomes broad, and its shape is symmetrical.

## 4. Results and Discussion

### 4.1. TEM

To visibly exhibit the filler particles on the rubber matrix, Figure 1 presents transmission electron microscopy (TEM) images of neat NBR, NBR-H20, NBR-H60, NBR-S20, and NBR-S60 samples. Comparing the TEM images of neat NBR and NBRs with fillers, we see that the shape and distributions of the filler were identified. NBR-H20 and NBR-H60 had spherical island shapes with particle sizes of 37 ± 6 nm. In NBR-S20 and S60, the size of the silica particles was 27 ± 8 nm. Such particles were distributed as partially condensed aggregates. Specifically, a prominent feature in NBR-H60 is the formation of a percolation channel or network. The shape of the particles is important for electrical conductivity to reach percolation, which requires a channel or network to be formed, such as for NBR-H60. The aspect ratio is defined by the ratio between the horizontal and vertical lengths of the particles. For instance, the larger the aspect ratio, the thinner and longer the particle shape of the filler, and the more advantageous it is to build the percolation path, even if a small amount of loading is added. In the case of NBR-H20 and NBR-H60 with spherical shapes, the aspect ratio was determined to be 1; this type of filler particle has the disadvantage of building a percolation path compared to particles that have a large aspect ratio. However, a channel can be formed by increasing the filler concentration, such as NBR-H60, for reaching percolation [33].

### 4.2. Impedance Spectroscopy at Room Temperature

The deconvolution results using self-developed analysis program (Appendix A) obtained for the measured dielectric loss spectra for one neat NBR and nine NBR-H, M, and S series at 298 K are summarized in Figure 2. From the deconvolution results, we recognize that all specimens had three main relaxation processes, assigned as α and α′ processes and DC conductivity. Figure 2a–c shows the evaluation results for the NBR-H, M, and S series, respectively. The open squares in Figure 2 represent the measured data for the dielectric loss. The simulated sum is represented by a black solid line, which consists of the contributions of the α and α′ processes (red and black dashed line HN function, respectively) and DC conductivity (purple dashed line). The distinctive feature for the spectrum of the α process is a broad frequency bandwidth (approximately six frequency decades on a logarithmic scale) and an asymmetric shape for a vertical line passing through the central frequency. However, the α′ process has a comparatively narrow frequency bandwidth (approximately four frequency decades) and symmetric shape for a vertical line passing through the central frequency. The central frequencies of the α and α′ processes normally appeared near 10^6^ Hz and 1 Hz, respectively. As mentioned in Section 3, the α process mainly originates from the rotational motion of the segmental chain under the external electric field. The α′ process originates from the flipping motion of the end-to-end vector of polymer segments in polymer type A. Meanwhile, DC conductivity exhibited a dominant contribution in the low-frequency band, which had a dominant contribution up to 10^4^ Hz for the neat NBR samples investigated in this study. The distribution of the DC conductivity originates from the migration effect of charge carriers.

From Figure 2a–c, we see that the central frequency of each relaxation process varied with filler loading at room temperature. In the NBR-H and M series, the central frequency of the α′ process indicated by arrows was shifted toward the high-frequency side with increasing CB filler concentration, together with increasing conductivity contributions, as indicated by an increase in the area. However, the NBR-S series shifted the central position of the α′ process to the low-frequency region with increasing silica filler concentration. The shape parameters of the α′ process for neat NBR were approximately *a* = 0.05 and *b* = 1. In the α process for neat NBR, the shape parameters were *a* = 0.48 and *b* = 1. In NBRs with fillers, the shape parameters were similar. Furthermore, the contribution of the conductivity and the position of the central frequency to the α′ process was influenced by the filler.

The influence of the central frequency for the α process in the NBRs with different filler types and contents on the filler is illustrated in Figure 3. In Figure 3, the central frequency is displayed as a function of filler concentration with uncertainty. In all cases, the central frequency of the α process in the NBR-H, M, and S series scarcely varied with filler type and loading. Likewise, Vieweg et al. [34] reported a similar observation to our study: negligible modification of the central frequency for the α process in styrene-butadiene rubber (SBR) with CB and silica filler. However, they did not mention the main reason for the negligible influence of the addition of filler on the α process. Similar observations were found in other studies in that the α process did not depend on filler particles, such as carbon nanotubes, zirconium dioxide, carbon black and silica, and polymer composites [35,36,37].

The variations in relaxation frequency for the α′ process in NBRs with different filler types and contents are displayed in Figure 4, together with the slope of the linear fit between f_0_ and filler content. The central frequency of the α′ process in the NBR-H and M series shifted toward the higher frequency side with increasing filler loading. Moreover, the shift rate for the relaxation frequency for the NBR-H series was 10^4^ times more significantly influenced by filler content than that for the NBR-M series. It can be explained from the percolation caused by HAF fillers, which is discussed later. Since the particle sizes of the HAF fillers were smaller than those of the MT fillers, the physical crosslink structures of the HAF composites were densely developed. As a result, the molecular mobility of the NBR-H series was significantly decreased compared with that of the MT composites. Therefore, we observed that the central frequency of the α′ process was highly influenced by the size of the fillers. However, the central frequency for the NBR-S series shifted toward the lower frequency region with increasing filler loading.

This behavior in the NBR-S series can be understood by zero shear viscosity as follows. One of the characteristic features of the dielectric relaxation process of the type-A polymer is the linear dependence of the relaxation time on the zero-shear viscosity (*η*). The general relation between relaxation time and zero-shear viscosity in a non-entangled bulk polymeric system can be written as follows [18]:(7)$$\tau_{p}=\frac{12M\eta }{\pi^{2}\rho RTp^{2}}$$
where η is the zero-shear viscosity, *M* is the molecular weight, ρ is the density, and p is the *n*-th order of the normal mode relaxation. According to Equation (7), the relaxation time was directly proportional to the zero-shear viscosity of a specimen. In prior research [38], the zero-shear viscosity of a rubber matrix was found to increase with increasing silica concentration. Thus, there was obvious evidence for a decrease in the central frequency for the NBR-S series. In addition, since silica filler also has an electric nonpolar property, the transport phenomenon for charge carriers in polymeric systems will be restricted under an external field. This can be another reason why the relaxation frequency shifts toward the low-frequency side with increasing nonpolar filler content.

The DC conductivity was obtained from the fitting results using Equation (6). The variation in DC conductivity as a function of the HAF, MT, and silica filler loading is shown in Figure 5. The magnitude of the DC conductivity for the NBR-H and M series increased with increasing filler content. However, the DC conductivity for the NBR-S series decreased with increasing filler content. The behavior was similar to the variation in the central frequency for the α′ process, as shown in Figure 4. In the NBR-H series of Figure 5, the red solid line indicates the fitting result by the described model as follows [39]:(8)$$\sigma_{dc}=\sigma_{f}{\left(v-v_{c}\right)}^{q},\;(v>v_{c})\;$$
where $\sigma_{dc}$ is dc conductivity of the composite, $\sigma_{f}\;$is the value of filler conductivity. v, $v_{c}$, and q indicate the filler concentration (loading), percolation threshold, and scaling exponent, respectively. Calculated percolation threshold $v_{c}$ was found to be 37 ± 1 phr for the NBR-H series, whose value was obtained by the fitting result. Likewise, in the NBR-H series of Figure 5, the red solid line shows a fitting result ($v_{c}$ = 41 ± 1 phr) using Equation (8). The obtained percolation thresholds for two fitting results showed similar values. In the case of the NBR-H and M series, since the CB filler has a positive polarity, the transport mechanism for the charge carrier was activated. Consequently, the average DC conductivity of the polymeric system increased with increasing CB loading. However, as mentioned before, since silica filler has a nonpolar property, an external electric field will be shielded. Therefore, the transport phenomenon for the charge carriers will be restricted, and the total DC conductivity in the polymeric system decreased with increasing silica filler content.

Furthermore, the magnitude of the DC conductivity for the NBR-H series was approximately 10^5^ times more rapidly varied than that for the NBR-M series. This can be understood by the close correlation between percolation and filler particle size. Since HAF has a smaller particle size (28~36 nm) than MT (250~350 nm), the percolation path for the NBR-H series was densely developed compared to the M-series. This observation was consistent with the TEM result. As a result, the densely developed percolation path can support the transport mechanism for the charge carrier, which acts as a conductive metal wire. In other words, a small HAF filler is crucial to percolation formation, leading to a drastic increase in conductivity.

### 4.3. Temperature Dependence of the Dielectric Relaxation

The dielectric properties of the NBRs with CB and silica fillers were investigated in the temperature range from 233 K to 403 K. The measured complex permittivity spectra were analyzed by a modified dispersion analysis program. From this deconvolution process, the central frequency determined for the α′ and α processes and DC conductivity for neat NBR versus reciprocal temperature is depicted in Figure 6. The DC conductivity and α′ process were mainly observed in the temperature range from 265 to 403 K, and the α process appeared in the temperature range from 248 to 303 K. The central frequencies for the α′ and α processes showed a nonlinear dependence for the logarithmic f_o_ and the reciprocal temperature, while the DC conductivity below 370 K showed a linear variation.

As the related dynamics of the polymer chain resulted in a sudden change near T_g_, the central frequency of the α process had a stronger temperature dependence than the α′ process and DC conductivity. From this observation, we can qualitatively infer that the α process had higher activation energy than the other two relaxation processes. Meanwhile, the value for the DC conductivity obtained for neat NBR lay in the range of 10^−6^ to 10^−10^ S/m, and these values were reasonably comparable to those found for other insulating rubber polymers in the literature [40,41,42,43].

Likewise, the central frequency of the α′ and α processes and DC conductivity versus reciprocal temperature in NBR-H20, M20, and S20 are representatively shown in Figure 7. The overall temperature dependence of all relaxation processes for NBRs on filler content shows behavior similar to that observed for neat NBR. The central frequencies of the α process were mainly distributed in the frequency range between 10^−3^ and 10^6^ Hz, and the α′ relaxation process occurred at approximately 10^−3^–10^3^ Hz. The DC conductivity was distributed at approximately 10^−10^–10^−6^ S/m.

Figure 8 represents the temperature dependence of the central frequency for the α process, as determined from the results of the deconvolution procedure for the dielectric loss spectra. The α process commonly appeared in the temperature range of 248–303 K, except for NBR-H40 and H60. In the case of the highest filler loading, the relaxation rate for segmental dipoles increased with decreasing segmental mobility because of the increased physical crosslink density according to the increased filler loading. Thus, the α relaxation process for NBR-H40 and 60 could not be observed in monitored frequency ranges (10^−2^–10^6^ Hz) [5]. Moreover, in the low-frequency region, since the contribution of DC conductivity rapidly increased with increasing HAF content, the spectra of the α process were hidden in the loss spectra of DC conductivity. Therefore, the deconvolution processes of α process for NBR-H40 and H60 could not be conducted.

In principle, T_g_ is empirically defined as the temperature at which the central frequency described by the VFTH function of Equation (2) reached 1.6 mHz, as shown in solid lines of Figure 8a. The T_g_ in the α process can be obtained by fitting the relaxation rate for the α process using the VFTH temperature dependence model. In this measurement, due to the lack of sampling points for the α process near T_g_ (250 K), we conducted the calculation on the basis of an extrapolation of the fitting results for the VFTH model. The results obtained from a fit of the VFTH model to the relaxation rate of the α process are depicted as a solid line in Figure 8a.

Meanwhile, the T_g_ by DSC measurement in rubber polymer was difficult to define as a single temperature, because of the wide distribution of T_g_ in the transition region. However, for comparison with dielectric relaxation spectroscopy (DRS), we tried to define T_g_ as a single temperature at which the slope of the tangent line in the heat flow curve had the maximum value between the glass transition regions scanned by DSC analyzer programs. According to the definition, the T_g_ was obtained through the DSC measurement.

The T_g_ values obtained using a fitting simulation based on the VFTH model by DRS and DSC were compared in Table 3. The T_g_ obtained for the NBRs with different filler types and contents using the two methods was found to be approximately 250 K. The obtained T_g_ value obtained for neat NBR by the DRS method was 250.0 ± 6.5 K. The T_g_ values obtained for neat NBR from DSC measurements were 250.1 ± 0.2 K. Likewise, the T_g_ for the NBRs with different filler types and contents was measured using DRS and DSC to be in the range of 250.1 ± 2.5 K and 248.4 ± 1.0 K, respectively. The T_g_ values obtained by the two different methods had a discrepancy of less than 4 K. Consequently, we found that the DRS and DSC results showed good agreement with each other. However, significant variations in the T_g_ values for different filler conditions were not found. Vieweg et al. [34] conducted an investigation into the influence of filler composites on T_g_ for reinforced SBR rubber using various CB and silica fillers. They also reported negligible variation in the T_g_ of reinforced SBR.

Because the glass transition of polymer generally occurred in the broad temperature range, it is difficult to evaluate the influence of CB and silica fillers on glass transition. Thus, the heat capacity change (ΔC_p_) at the glass transition region was obtained from the DSC heat flow curves divided by heating rate (K/min) in the region of glass transition (Appendix A). The ΔC_p_ values of NBRs were found to be decreased with increasing the content of CB and silica fillers (Figure 9). The ΔC_p_ value was strongly related to the volume of the immobilized rubber chains [44]. The decrease in the mobility of the polymer chain is mainly caused by the interaction between rubber chains and surface of the filling agents [45,46]. As a consequence, the ΔC_p_ values of NBRs decrease with increase in the volume fraction of the filling agents. In addition, Mostafa et al. [47] showed the similar result that represented the ability of CB in NBR to impart greater stiffness to the filled vulcanizates, which reduces the mobility of the rubber chains.

The central frequency of the α′ process versus reciprocal temperature is shown in Figure 10. The α′ process generally appears in the temperature range of 268–403 K, except for NBR-H40 and 60. Similar to Figure 8, since the contribution of DC conductivity to the permittivity spectra rapidly increased with increasing HAF filler content, the morphologies of the α′ process were buried due to the contribution of DC conductivity. Therefore, the fitting procedures in this process could not be conducted.

The activation energy for the α′ process was obtained by the VFTH model Equation (2). The obtained activation energies versus filler content for the α′ process are displayed in Figure 11. The blue solid lines show the results obtained from a linear fit for the activation energy versus filler concentration for the M series. From the slope of the linear fitting result for the M series, we see that the activation energies slightly decreased with increasing filler content, while the activation energies of the NBR-S series significantly increased with increasing filler content. This result shows the opposite tendency in the shift behavior for the central frequencies, as shown in Figure 4. In the case of the NBR-S series, the central frequency shifted toward the low-frequency side with increasing filler content, as shown in Figure 4, which is by the variation in activation energy. This means that the motion of the end-to-end vector became interrupted by filler loading, resulting in an increase in the activation energy. The NBRs with CB had a smaller effect on the activation energy than NBR with silica.

Figure 12 shows the temperature dependence of the DC conductivity for neat NBR and NBR-H and the M and S series samples. The activation energy for the DC conductivity can be obtained by fitting the Arrhenius temperature dependence in Equation (5). The solid lines indicate the results of the Arrhenius fitting, and the obtained activation energies are illustrated in Figure 12. The fitting procedure was only performed for the experimental results obtained from 363 to 293 K because the fitting results above 363 K deviate from an Arrhenius temperature dependence. The DC conductivity in this temperature region abnormally decreased with increasing temperature. Similar behavior has already been observed in analogous polymeric systems [48,49,50,51,52]. Aziz et al. [51] reported that the main reason for these abnormal behaviors is a phase transition from the crystallized amorphous phase to only the amorphous phase of the polymer host. In addition, Roggero et al. [52] proposed that the decrease in conductivity with increasing temperature is governed by the outgassing process of humid air from polymeric systems. The abnormal behavior observed in this work was strongly influenced by the filler type and content. In the case of CB filler composites (Figure 12), the transition temperature for the abnormal behavior increased with increasing filler content. In contrast, for the case of silica filler composites, the transition temperature inversely decreased with increasing filler content.

The obtained activation energies for DC conductivity versus filler content for the H, M, and S series are plotted in Figure 13. For comparison with our result, the activation energies determined for NBRs with HAF by the Yehia Group [53] were indicated as the black-filled circles. The slopes for the four solid lines indicate the results obtained from a linear fit for E_a_ versus filler content. The activation energies for the NBR-S series showed little change with increasing filler content. However, the activation energies for the NBR-H and M series decreased with increasing filler content. Furthermore, the activation energies for the NBR-H series rapidly decreased with increasing filler content compared with the M series. The rates of decrease in the activation energies are the same within the uncertainty as those reported in [53]. This tendency is also similar to the variation in DC conductivity shown in Figure 5. This phenomenon may arise from the fact that the charge carriers were decoupled from the segmental motion of the polymer chains and the fact that transport occurred via an activated hopping mechanism [54].

## 5. Conclusions

The influences of carbon black and silica fillers with different concentrations on the dielectric properties of NBR co-polymer were investigated by using dielectric relaxation spectroscopy (DRS) with a self-developed analysis program. Mainly, the α and α′ relaxation process and contribution of the DC conductivity were observed in all NBR specimens. The observed complex dielectric spectra were fitted by using the Havriliak–Nagami model and conductivity contribution. The temperature dependencies of the α and α′ relaxation and DC conductivity followed VFTH and Arrhenius behavior, respectively.

The central frequencies of α′ relaxation were shifted toward the higher and lower frequency regions with increasing the carbon black and silica fillers, respectively. Moreover, the DC conductivities of NBR were markedly affected by the addition of fillers. Conspicuously, the DC conductivity of the NBR-H series (HAF, N330) was rapidly varied with the addition of HAF fillers. It has been pointed out that the HAF fillers have a great influence on increasing the DC ionic conduction in NBR co-polymer because of the percolation effect. Therefore, the calculation of the percolation threshold using the Equation (8) has been conducted on both α′ relaxation and DC conductivity of NBR-H series, and the threshold points were obtained to be 37 and 41 phr, respectively.

Using the DRS and DSC methods, we measured the glass transition temperature (T_g_) of all NBR specimens to be in the range of 250.1 ± 2.5 K and 248.4 ± 1.0. The obtained T_g_ using two different methods had good agreement with each other. Furthermore, the heat capacity change (ΔC_p_) values were obtained to evaluate the influence of filling agents on the glass transition. Although the noticeable influences of CB and silica fillers on T_g_ were not found, the results of ΔC_p_ values were clearly revealed in that the ΔC_p_ in NBRs decreased with increasing the CB and silica filler. From these results, we identified that the addition of filling agents leads to a decrease in the mobile rubber chains of NBR.

## Figures and Tables

**Figure 1 polymers-14-00155-f001:**
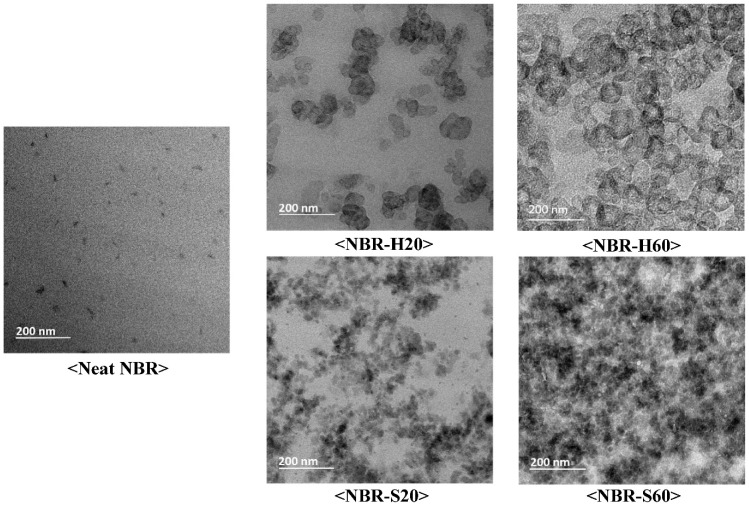
TEM image of neat NBR, NBR-H20, NBR-H60, NBR-S20, and NBR-S60.

**Figure 2 polymers-14-00155-f002:**
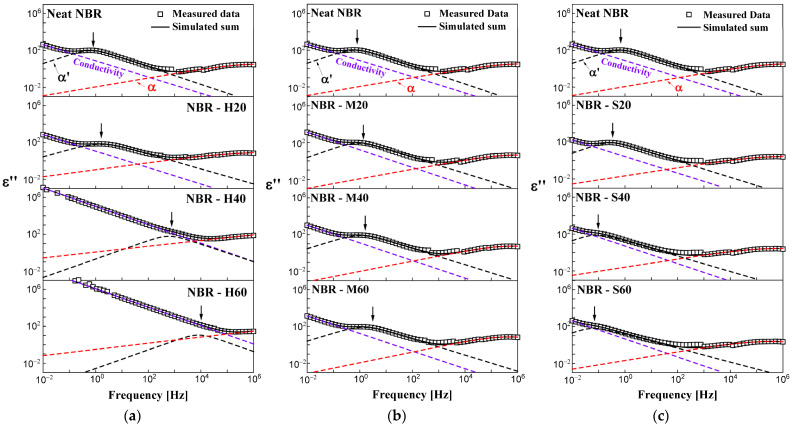
Influence of filler loading on the dielectric loss spectra in the frequency range from 0.01 Hz to 1 MHz for neat NBR and (**a**) NBR-H series, (**b**) M series, and (**c**) S series at 298 K. To compare neat NBR and NBRs with filler, we show here three identical neat NBR loss spectra on the top side of each figure (**a**–**c**). The arrows indicate the central frequency for the α′ process.

**Figure 3 polymers-14-00155-f003:**
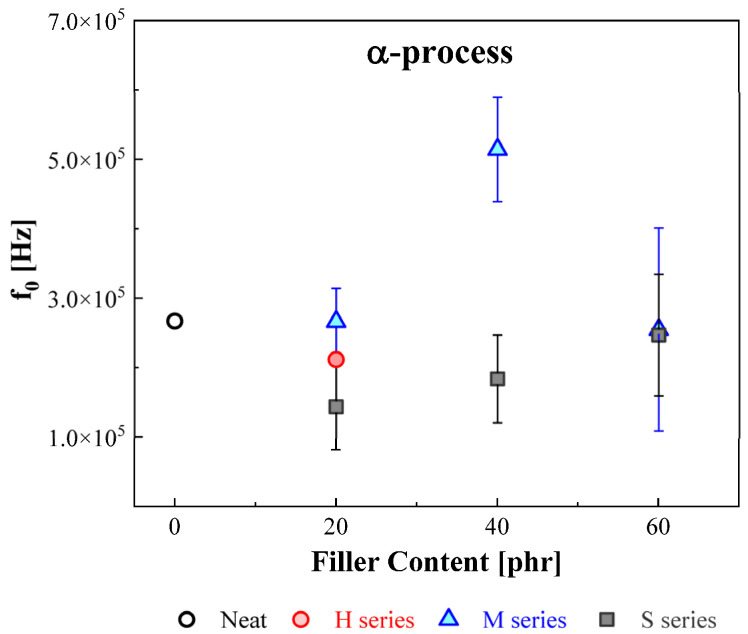
Variation of the central frequency for the α process in NBRs with different filler types and contents. The red circle indicates the H series, and the blue triangle and gray square symbols indicate the M and S series, respectively.

**Figure 4 polymers-14-00155-f004:**
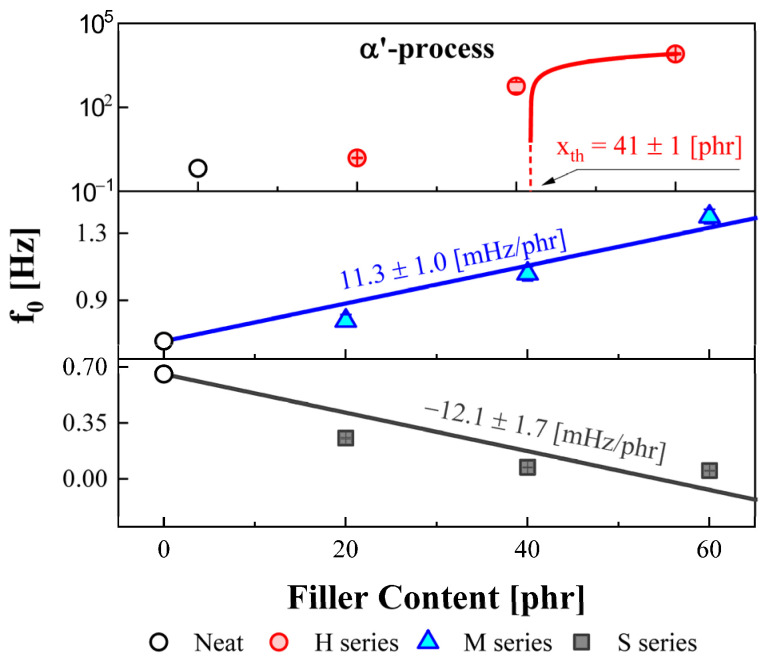
Variation of the central relaxation frequency for the α′ process with different filler types and contents. The red circle indicates the H series, and the blue triangle and gray square symbols indicate the M and S series, respectively. The red solid line indicates that fitting result of Equation (8) and blue and gray solid lines represent the linear fitting results of f0 versus the filler concentration.

**Figure 5 polymers-14-00155-f005:**
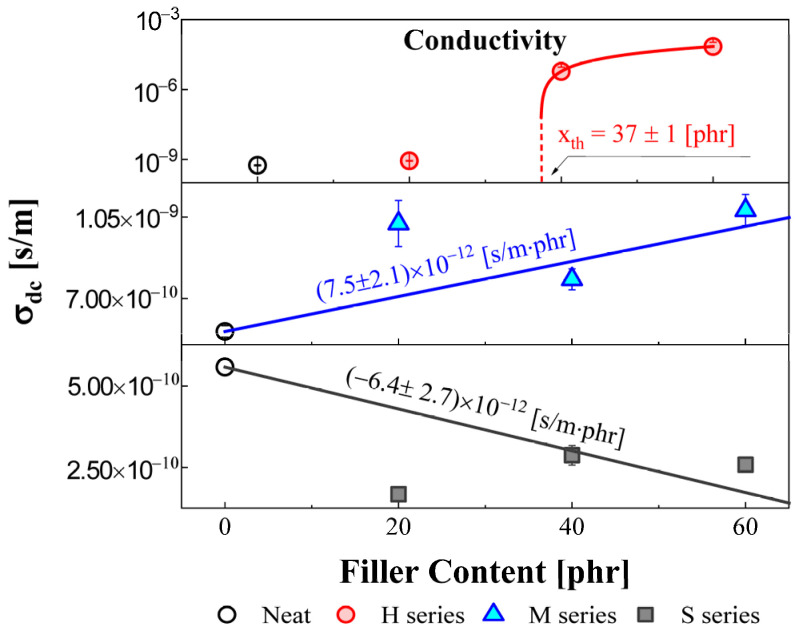
Conductivity for NBRs with different filler types and contents at 298 K. The red circle indicates the H series, and the blue triangle and gray square symbols indicate the M and S series, respectively. The red solid line indicates the fitting result of Equation (8).

**Figure 6 polymers-14-00155-f006:**
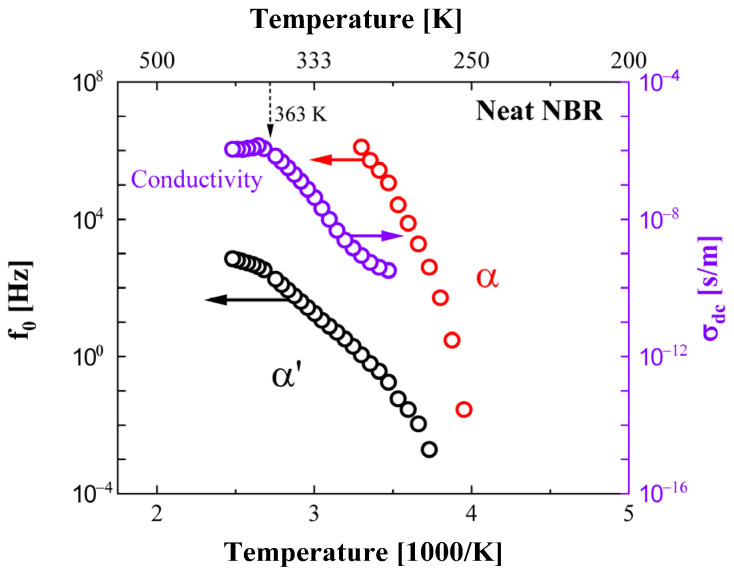
The variation of the central frequency for the α′ and α relaxation processes and DC conductivity versus reciprocal temperature for neat NBR. The black circles indicate the α′ process, and the violet and red circles indicate the DC conductivity and α process, respectively.

**Figure 7 polymers-14-00155-f007:**
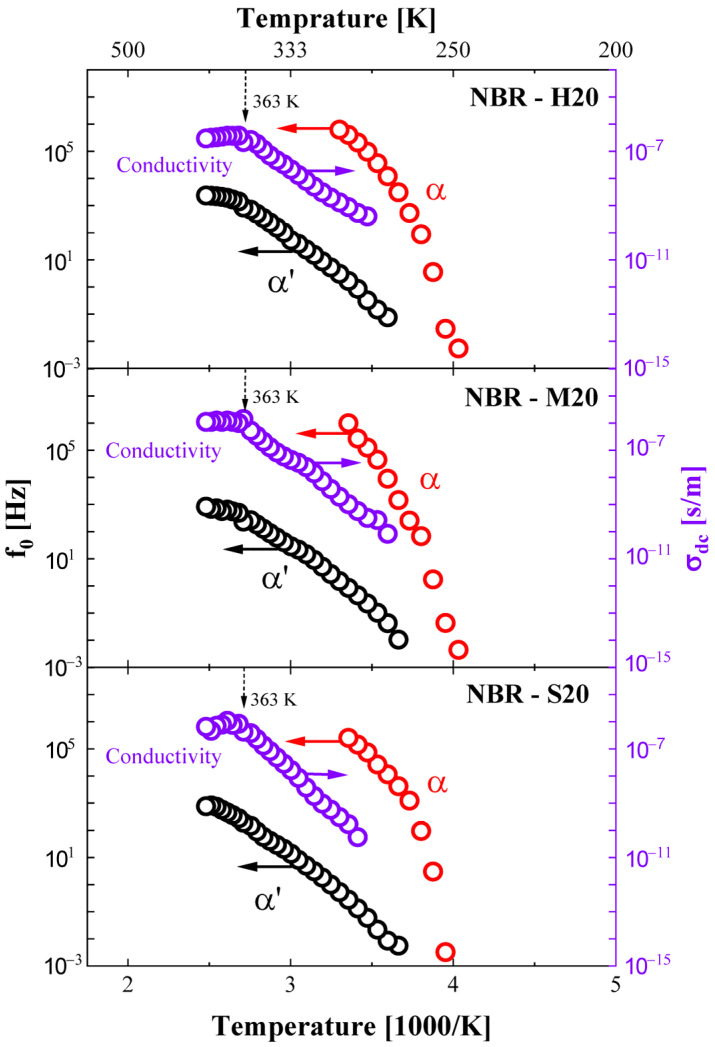
The variation of the central frequency for each relaxation process and DC conductivity as a function of reciprocal temperature for NBR-H20, NBR-M20, and NBR-S20. The black circles indicate the α′ process, and the violet and red circles indicate the DC conductivity and α process, respectively.

**Figure 8 polymers-14-00155-f008:**
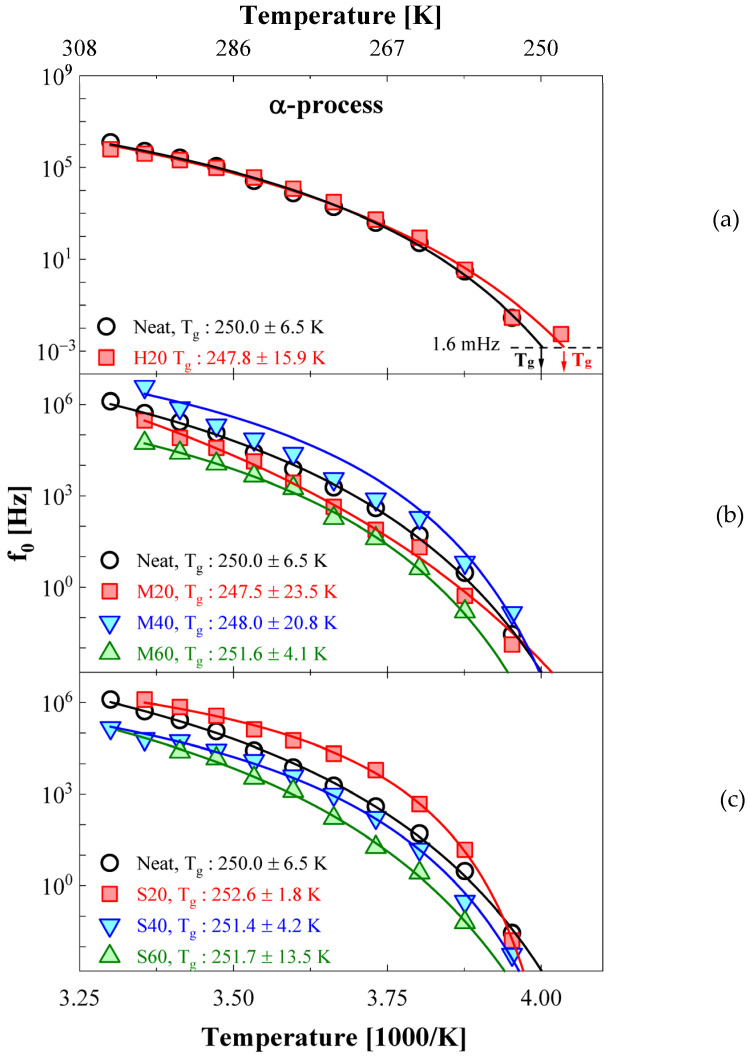
Temperature dependence of the central frequency for the α process on different fillers and contents. (**a**–**c**) Each H, M, S series. In (**b**,**c**), f0 is multiplied by arbitrary values so that it can be better distinguished by upshifting or downshifting on the frequency axis. For comparison, three identical neat NBR results are shown on the top, middle, and bottom sides of each figure.

**Figure 9 polymers-14-00155-f009:**
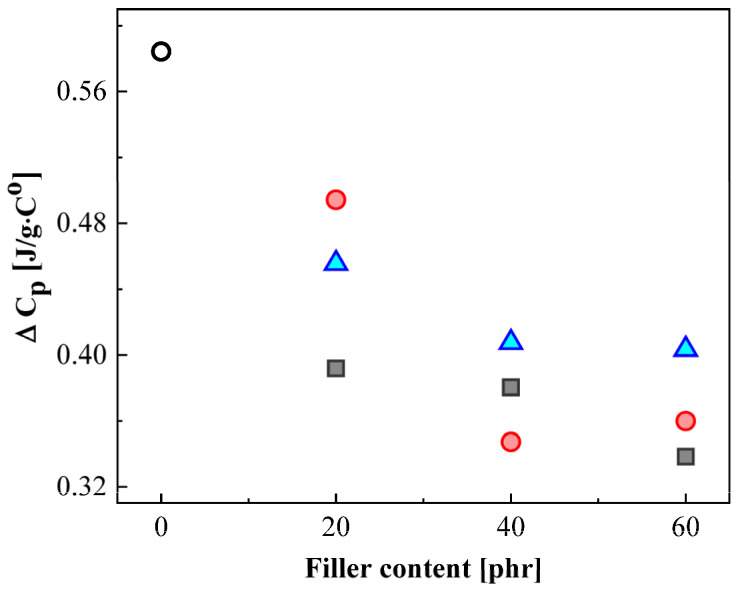
The summarization of obtained Delta C_p_ values as a function of filler content. The empty black circle indicates the result of neat NBR, and the black square, red circle, and blue triangle symbols represent the results of NBR with HAF, MT, and silica fillers, respectively.

**Figure 10 polymers-14-00155-f010:**
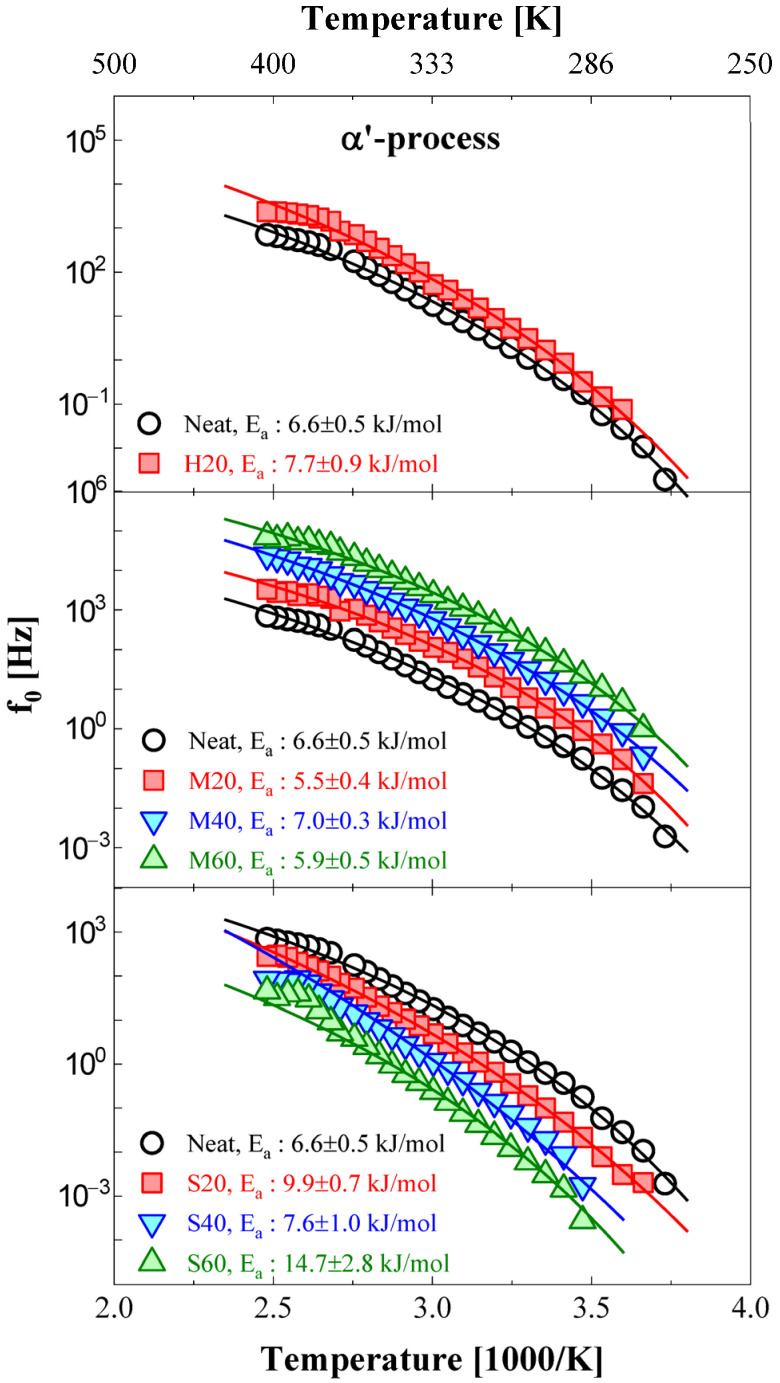
Temperature dependence of the relaxation rate for the α′ process on different filler loading contents and types. The f_0_ multiplied by arbitrary values can be better distinguished by downshifting on the frequency axis. For comparison, three identical neat NBR results are shown on the top, middle, and bottom sides of each figure.

**Figure 11 polymers-14-00155-f011:**
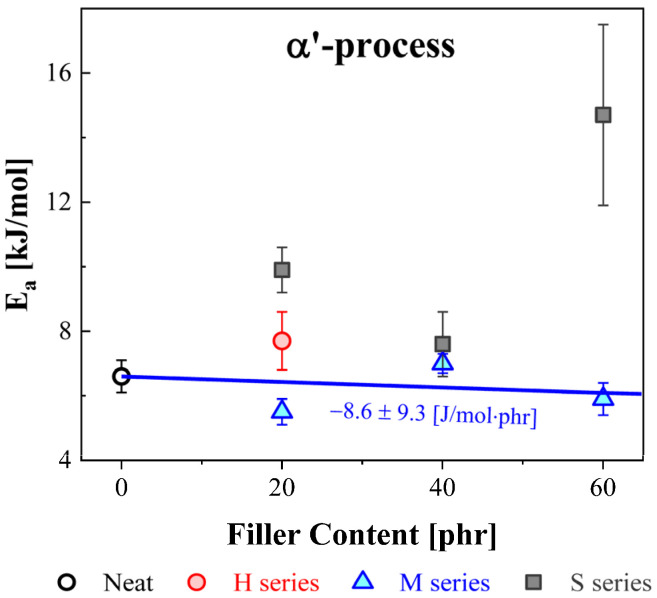
Comparison of activation energy with the different filler loading contents and types. The blue solid lines represent the linear fitting results for the M series.

**Figure 12 polymers-14-00155-f012:**
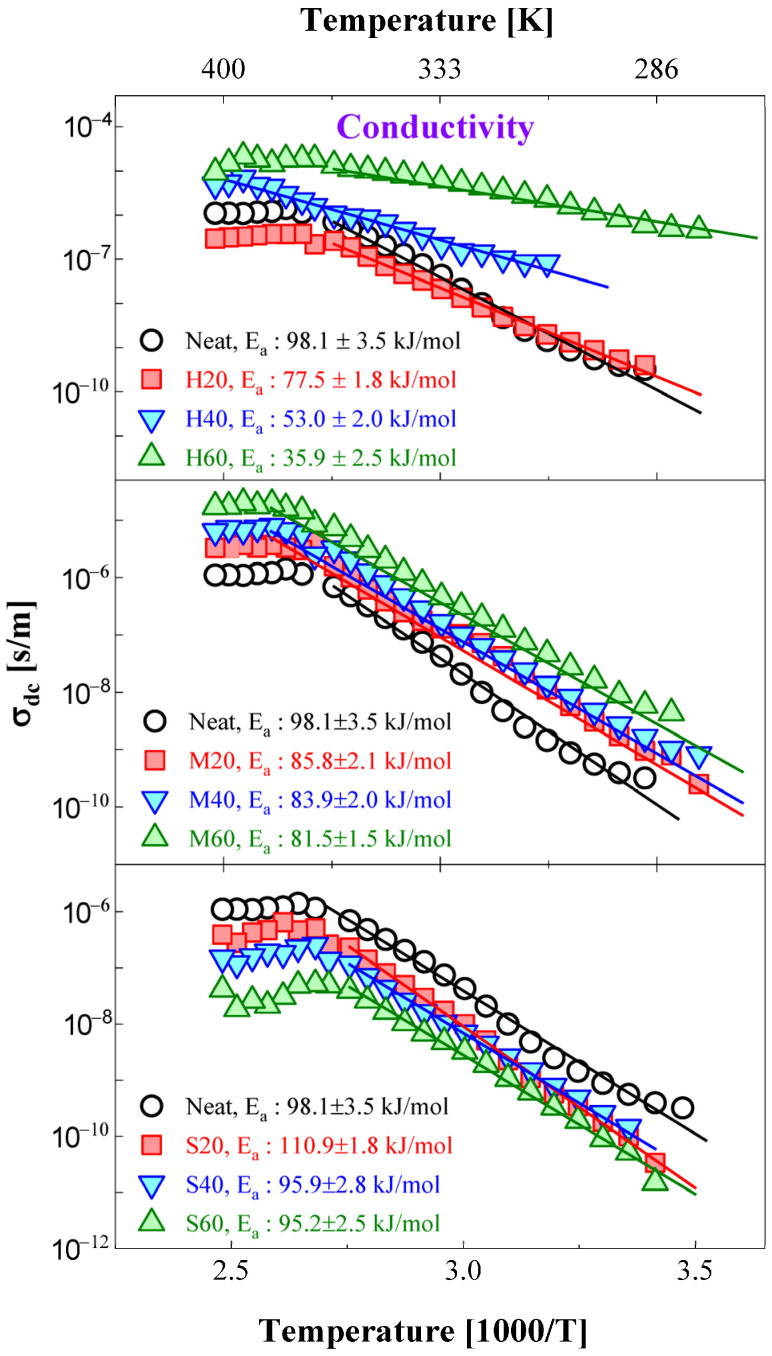
Temperature dependence of DC conductivity for NBR blends with different filler types and contents. The activation energies for each conduction process were determined to show an Arrhenius temperature dependence, as represented by the solid line. The conductivity data multiplied by arbitrary values can be better distinguished by downshifting on the conductivity axis. For comparison, three identical neat NBR results are shown on the top, middle, and bottom sides of each figure.

**Figure 13 polymers-14-00155-f013:**
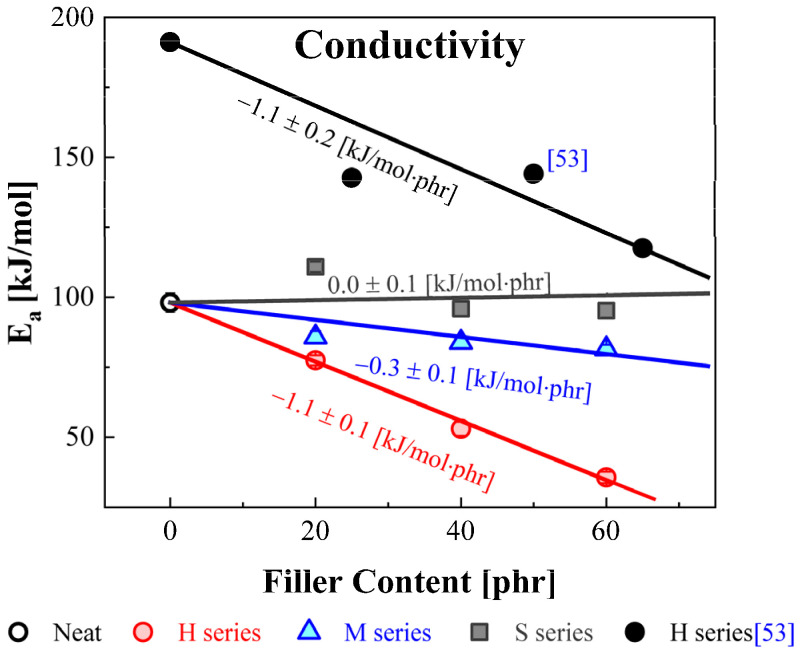
The activation energy for DC conductivity for NBR composites under varying filler loading content. The solid lines and their slopes represent the linear fitting results obtained for the H, M, and S series.

**Table 1 polymers-14-00155-t001:** Compositions of the NBR with HAF and MT CB fillers.

Composition	Neat NBR	NBR-H20	NBR-H40	NBR-H60	NBR-M20	NBR-M40	NBR-M60
KNB 35 L	100	100	100	100	100	100	100
ZnO	3.0	3.0	3.0	3.0	3.0	3.0	3.0
St/A *	1.0	1.0	1.0	1.0	1.0	1.0	1.0
HAF N330	-	20	40	60	-	-	-
MT N990	-	-	-	-	20	40	60
S	1.5	1.5	1.5	1.5	1.5	1.5	1.5
TBBS ^+^	0.7	0.7	0.7	0.7	0.7	0.7	0.7

* St/A: stearic acid; ^+^ TBBS: N-tert-butyl-2-benzothiazole sulfenamide.

**Table 2 polymers-14-00155-t002:** Compositions of the NBR with silica fillers.

Composition	NBR-S20	NBR-S40	NBR-S60
KBR 35 L	100	100	100
ZnO	3.0	3.0	3.0
St/A	1.0	1.0	1.0
Silica S-175	20	40	60
Si-69 ^×^	1.6	3.2	4.8
PEG ^#^	0.8	1.6	2.4
S	1.5	1.5	1.5
TBBS ^+^	0.7	0.7	0.7

^×^ Si-69: silane coupling agent; ^#^ PEG: polyethylene glycol; ^+^ TBBS: N-tert-butyl-2-benzothiazole sulfenamide.

**Table 3 polymers-14-00155-t003:** Summary of the T_g_ values obtained with DRS and DSC.

	Neat NBR		NBR H Series		NBR M Series		NBR S Series	
Contents (phr)	DRS (K)	DSC (K)	DRS (K)	DSC (K)	DRS (K)	DSC (K)	DRS (K)	DSC (K)
0	250.0 ± 6.5	250.1 ± 0.2	-	-	-	-	-	-
20	-	-	247.8 ± 15.9	248.6 ± 0.2	247.5 ± 23.5	247.8 ± 0.2	252.6 ± 1.8	248.0 ± 0.1
40	-	-	-	249.2 ± 0.1	248.0 ± 20.8	248.8 ± 0.1	251.4 ± 4.2	249.6 ± 0.1
60	-	-	-	247.9 ± 0.1	251.6 ± 4.1	247.6 ± 0.1	251.7 ± 13.5	247.9 ± 0.1

## Data Availability

The data used to support the findings of this study are available from the corresponding author upon request.

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
