# Peer review of "Influence of Carbon Black and Silica Fillers with Different Concentrations on Dielectric Relaxation in Nitrile Butadiene Rubber Investigated by Impedance Spectroscopy"

_polymers, 2021, doi:10.3390/polym14010155_

Round 1

Reviewer 1 Report

The technical article focuses on the impedance spectroscopic study of the impacts of carbon black and silica fillers on the dielectric relaxation behavior of Nitrile rubber. Overall, the paper presents some intriguing results, however, there are some aspects that need to be addressed before it can get published.  A major revision of the manuscript is recommended.

Introduction Section:

In general, the Introduction should summarize any existing research already conducted on NBR in the same line of the paper to highlight the scope or the gaps. If the current work is unprecedented, authors should highlight that. The objective of the research is unclear from the manuscript and does not specify why it is relevant or important for the readers. 

  • Page-1, Line 41, the author comment: “…glass transition temperatures of rubber vulcanized are improved by strengthening with fillers such as carbon black…”. In reality, the filler content does not have an appreciable influence on the glass transition temperature. A strong polymer-filler interaction at times causes some minor shifts in glass transition temperatures, but more towards the positive side.
  • Page 2, Line 54 and 55: The references “silicone rubber vulcanizates” are irrelevant for this paper. There are numerous research publications available on the use of silica filler in nitrile rubber. Authors should refer to some of them.
  • Page 2, Line 60 and 63: What is the meaning of “polymeric rubber”? Rubbers are polymers anyway.
  • Page 2, Lines 73-75 are the conclusions and should be deleted from the introduction part.

Experimental Section:

  • The term “virgin NBR” typically indicates the raw rubber, without any additives to it. This is clearly not the case here, therefore, the authors should use a different terminology throughout the paper to represent the NBR compound without filler.
  • Headings of Table 1 and Table 2 may simply read as “Compositions of NBR Compounds….not “ “Chemical Composition”.
  • Line 87: The sentence “ the silica was produced by Zeosil 175…” is wrong. Zeosil 175 is a specific grade of precipitated silica that is produced by Solvay.

Results and Discussion Section:

  1. Line 194: “polarized particle sizes”- what does that even mean?
  2. Particle sizes are referred in the text as “few tens of nm”- authors should specify the average value range.
  3. Line 197: “The dense permeated H60 allows charge carriers to move easily”.. This is an observation from the impedance data. The authors therefore may choose to provide the Impedance data first and then TEM, as the supporting evidence.
  4. Line 208: The comment “silica particles are nonpolar” is wrong. The precipitated silica (Zeosil 175) has, in fact, strong surface polarity.
  5. Figure-2 is difficult to follow. Particularly the 3D plot, what information does it bring? Hard to visualize the red and blue trend lines. A plot like this may be included as a supplementary document if at all needed.
  6. Figure-3. The ‘Blue” and “purple” lines are almost impossible to differentiate and distinguish. Authors either should use distinctly dissimilar colors or use different legends/ line markers.
  7. Line 246: the term “symmetrical spectrum shape” should be clarified.
  8. Line 293 & 390: It should be clarified that the “crosslink structures” or “crosslink density” referred are the “physical crosslinks” not the chemical ones. Higher structure/ surface area of HAF contributes to enhanced polymer-filler interaction and thus causing reduced segmental mobility of the polymer chains.
  9. Line 394: The comment “morphologies of the process was buried…” is not clear to understand.
  10. Line 435: NBR is an amorphous polymer. So the argument on the decrease of its “crystallinity” upon the addition of fillers is not correct. The influence of fillers in NBR on DCp or Tg should be correlated to the rubber-filler interaction.
  11. Line 497: Why the result from a different research group is included in Figure-14? The authors should provide some solid justification.

Author Response

Dear Reviewer,

We really appreciate your valuable comments to improve the quality of our manuscript.

Followings are collected in response to your comments.

Comments for the author

The technical article focuses on the impedance spectroscopic study of the impacts of carbon black and silica fillers on the dielectric relaxation behavior of Nitrile rubber. Overall, the paper presents some intriguing results, however, there are some aspects that need to be addressed before it can get published. A major revision of the manuscript is recommended.

Introduction Section:

In general, the Introduction should summarize any existing research already conducted on NBR in the same line of the paper to highlight the scope or the gaps. If the current work is unprecedented, authors should highlight that. The objective of the research is unclear from the manuscript and does not specify why it is relevant or important for the readers. 

   Answer: In response to reviewer’s comment, the objective and motivation of this research are described in the introduction (line 61~66, 70~72).

Page-1, Line 41, the author comment: “…glass transition temperatures of rubber vulcanized are improved by strengthening with fillers such as carbon black…”. In reality, the filler content does not have an appreciable influence on the glass transition temperature. A strong polymer-filler interaction at times causes some minor shifts in glass transition temperatures, but more towards the positive side.

Answer: In response to reviewer’s comments, we replaced “and glass transition temperature” with “chemical stability and heat resistance” in line 41. The related descriptions, reference [8] is inserted in text and reference.

Page 2, Line 54 and 55: The references “silicone rubber vulcanizates” are irrelevant for this paper. There are numerous research publications available on the use of silica filler in nitrile rubber. Authors should refer to some of them.

Answer: We changed “silicone rubber vulcanizates” to “NBR rubber vulcanizates”. In addition, we included references 15 and 16 that contain the NBR researches instead of silicone rubber-related references (line 54-55).

Page 2, Line 60 and 63: What is the meaning of “polymeric rubber”? Rubbers are polymers anyway.

Answer: In response to reviewer's comment, we revised “polymeric rubber” to “rubber” or “NBR” in the revised manuscript (line 59, 67). 

Page 2, Lines 73-75 are the conclusions and should be deleted from the introduction part.

Answer: As you mentioned, we deleted the sentence from the introduction.

 Experimental Section:

The term “virgin NBR” typically indicates the raw rubber, without any additives to it. This is clearly not the case here, therefore, the authors should use a different terminology throughout the paper to represent the NBR compound without filler.

Answer: In response to reviewer’s comment, we replaced “virgin” to “neat” to represent the NBR without filler. Polymer industry commonly use “Neat Polymer” to represent the purest form of polymer.

Headings of Table 1 and Table 2 may simply read as “Compositions of NBR Compounds….not “ “Chemical Composition”.

Answer: We deleted “Chemical” from “Chemical Composition” according to reviewer’s comment (line 83, 112, 114).

Line 87: The sentence “The silica was produced by Zeosil 175…” is wrong. Zeosil 175 is a specific grade of precipitated silica that is produced by Solvay.

Answer: We agree with reviewer’s comment. Thus, we revised as “The silica (specific grade Zeosil® 175) was produced by Solvay” (line 89, 90).

Results and Discussion Section:

Line 194: “polarized particle sizes”- what does that even mean?

Answer: In response to reviewer’s comment, “polarized” is wrong. Thus we removed the word “polarized” (line 197).

Particle sizes are referred in the text as “few tens of nm”- authors should specify the average value range.

Answer: As commented by the reviewer, we inserted the average value for particle sizes of the filling agent with the uncertainty. To obtain the particle size of a filling agent in the TEM image, we measured the size of 20-grain forms in each type of filler. Thus, we could obtain the average and uncertainty values of the particle size. The obtained size and uncertainty of HAF and silica filler were inserted in the revised manuscript (line 196-198).

Line 197: “The dense permeated H60 allows charge carriers to move easily”.. This is an observation from the impedance data. The authors therefore may choose to provide the Impedance data first and then TEM, as the supporting evidence.

Answer: We removed the phrase “The dense permeated H60 allows charge carriers to move easily” because this description is not appropriate in this TEM section.

Line 208: The comment “silica particles are nonpolar” is wrong. The precipitated silica (Zeosil 175) has, in fact, strong surface polarity.

Answer: The sentence “However, although NBR-S60 has a percolation path, silica particles are nonpolar, leading to a decrease in conductivity.” is not appropriate in this TEM section. Thus, we removed this sentence from the TEM section.

Figure-2 is difficult to follow. Particularly the 3D plot, what information does it bring? Hard to visualize the red and blue trend lines. A plot like this may be included as a supplementary document if at all needed.

Answer: Figure 2 is included to demonstrate our simulation result using the dispersion analysis program. In response to the reviewer’s comment, we moves these figure and related contents to the supplementary materials. In supplementary materials, we also supplements the descriptions for 3 D plot.

Figure-3. The ‘Blue” and “purple” lines are almost impossible to differentiate and distinguish. Authors either should use distinctly dissimilar colors or use different legends/ line markers.

Answer: In response to the reviewer’s comment, we try to find the “Blue” line. However, we couldn’t find “blue” line in Figure 3.

Line 246: the term “symmetrical spectrum shape” should be clarified.

Answer: We agree with reviewer’s comment that this terminology is quite ambiguous. The “symmetrical spectrum shape” means symmetric shape in the low and high-frequency sides for a vertical line passing through the central frequency of the peak. Thus, we inserted the related explanation in the revised manuscript to avoid confusion (line 225-226).

Line 293 & 390: It should be clarified that the “crosslink structures” or “crosslink density” referred are the “physical crosslinks” not the chemical ones. Higher structure/ surface area of HAF contributes to enhanced polymer-filler interaction and thus causing reduced segmental mobility of the polymer chains.

Answer: In response to reviewer’s comment, we revised “crosslink..” to “physical crosslink…” in the revised manuscript (line 270, 366).

Line 394: The comment “morphologies of the process was buried…” is not clear to understand.

Answer: In response to reviewer’s comment, we revised this phrase to “The spectra of the α process were hidden in the loss spectra of DC conductivity” to make it clear (line 369-371).

Line 435: NBR is an amorphous polymer. So the argument on the decrease of its “crystallinity” upon the addition of fillers is not correct. The influence of fillers in NBR on DCp or Tg should be correlated to the rubber-filler interaction.

Answer: In response to the reviewer’s comment, we have reconsidered the fillers effects on ΔCp. Thus, we added the descriptions for heat capacity change (ΔCp) in views of rubber-filler interaction together with related references [44-47] (line 408-415).

Line 497: Why the result from a different research group is included in Figure-14? The authors should provide some solid justification.

Answer: To compare with our result, we included the data from Yehia Group [53]. Our results are found to be consistent with the data from Yehia Group.  

Reviewer 2 Report

The work «Improved performance of solid polymer electrolyte for lithium-metal batteries via hot press rolling» is actual and interesting. The introduction describes in detail the problem to be solved by the study, possible solutions to this problem, their advantages and disadvantages. The advantages of the proposed approach and selected materials are indicated. In section «Materials and Methods» all stages of the experiment are described in detail, supported by informative Figures in the form of diagrams and photographs. The results obtained are presented in the form of graphs with an explanation of the revealed patterns. The conclusion contains the values of key characteristics and summarizes the research carried out.

I have same comments:

Figure 2 needs to get rid of unnecessary details from the software.
Arrhenius plots should be more traditional.
In Figure 6, for the H series in the range from 20 to 40 phr, the conductivity changes by three orders of magnitude, it is obvious that the percolation threshold is located precisely in these filler concentrations.
Perhaps the authors can use a more classical model to calculate the percolation threshold, for example, Journal of Applied Polymer Science (2021): 51168.

Accept after minor revision

Author Response

Dear Reviewer,

We really appreciate your valuable comments to improve the quality of our manuscript.

Followings are collected in response to your comments.

The work «Improved performance of solid polymer electrolyte for lithium-metal batteries via hot press rolling» is actual and interesting. The introduction describes in detail the problem to be solved by the study, possible solutions to this problem, their advantages and disadvantages. The advantages of the proposed approach and selected materials are indicated. In section «Materials and Methods» all stages of the experiment are described in detail, supported by informative Figures in the form of diagrams and photographs. The results obtained are presented in the form of graphs with an explanation of the revealed patterns. The conclusion contains the values of key characteristics and summarizes the research carried out.

I have same comments:

Figure 2 needs to get rid of unnecessary details from the software.

Answer: Figure 2 is included to demonstrate our simulation result using the dispersion analysis program. In response to first reviewer’s comment, we moved these figure and related text to the supplementary materials, instead of removal of Fig. 2.

Arrhenius plots should be more traditional.

Answer: Because the fitting results (Fig. 12) above 363 K deviate from an Arrhenius temperature dependence and the DC conductivity in this temperature region abnormally decreased with increasing temperature, so we couldn’t perform the Arrhenius plot above the 363 K temperature region. We already described the related description in the revised manuscript as follows (line 453-461):

“The fitting procedure was only performed for the experimental results obtained from 363 K to 293 K because the fitting results above 363 K deviate from an Arrhenius temperature dependence. The DC conductivity in this temperature region abnormally decreased with increasing temperature. Similar behavior has already been observed in analogous polymeric systems [48-52]. Aziz et al. [51] reported that the main reason for these abnormal behaviors is a phase transition from the crystallized amorphous phase to only the amorphous phase of the polymer host. In addition, Roggero et al. [52] proposed that the decrease in conductivity with increasing temperature is governed by the outgassing process of humid air from polymeric systems.”.

In Figure 6, for the H series in the range from 20 to 40 phr, the conductivity changes by three orders of magnitude, it is obvious that the percolation threshold is located precisely in these filler concentrations.
Perhaps the authors can use a more classical model to calculate the percolation threshold, for example, Journal of Applied Polymer Science (2021): 51168.

Answer: In response to the reviewer’s comment, we conducted the recalculation of the percolation threshold value of NBR with HAF filler using the classical model suggested by reviewer. The percolation value was found to be 37 phr. We revised the percolation value with the obtained value of 37 phr and the related description in the revised manuscript. The revised contents are found in Figs. 4 and 5, Eq. (8) and line 304 to 306 of revised manuscript.

Accept after minor revision

We appreciate your valuable comments and thank you for your effort in the COVID-19 situation. We hope you stay healthy and happy.

Round 2

Reviewer 1 Report

Authors may consider some minor changes:

Page 1, L42: "..through incorporation.." may be more appropriate than " ...by strengthening..."

Page 2 L46: While the comment "...especially synergistic effects with nanotubes" may be technically correct, but not many industrial applications of rubber include the use of nanotubes anyway. On the other hand, the use of CB is far more prevalent due to its now cost and better dispersibility. Authors may relook at that.

Page 2, L 70: Rather than using "....based on these research data..." authors may consider mentioning something in the context of "...as a continuation of that line of research...".

Table 1&2: In place of "..NBR series...", just "..NBR.." would be sufficient.

Author Response

Dear Reviewer,

We really appreciate your valuable comments to improve the quality of our manuscript.

Followings are collected in response to your comments.

Comments for the Reviewer 1

Authors may consider some minor changes:

Page 1, L42: "..through incorporation.." may be more appropriate than " ...by strengthening..."

Answer : We agree with reviewer’s comment, we replaced “...by strengthening...” with “...through incorporation...” according to the comment.   

Page 2 L46: While the comment "...especially synergistic effects with nanotubes" may be technically correct, but not many industrial applications of rubber include the use of nanotubes anyway. On the other hand, the use of CB is far more prevalent due to its now cost and better dispersibility. Authors may relook at that.

Answer : In response to reviewer’s comment, the descriptions of the carbon nanotube have been deleted from the “introduction” to avoid confusion.

Page 2, L 70: Rather than using "....based on these research data..." authors may consider mentioning something in the context of "...as a continuation of that line of research...".

Answer : We modified the sentence “based on these research data” to “as a continuation of that line of research”.

Table 1&2: In place of "..NBR series...", just "..NBR.." would be sufficient.

Answer :“series” has been deleted in the captions of Table 1&2 according to the reviewer’s comment.